# Chemotherapy in Well Differentiated Neuroendocrine Tumors (NET) G1, G2, and G3: A Narrative Review

**DOI:** 10.3390/jcm12020717

**Published:** 2023-01-16

**Authors:** Arianna Zappi, Irene Persano, Linda Galvani, Elena Parlagreco, Elisa Andrini, Davide Campana, Maria Pia Brizzi, Giuseppe Lamberti, Anna La Salvia

**Affiliations:** 1Department of Specialized, Experimental and Diagnostic Medicine, S. Orsola-Malpighi Hospital, University of Bologna, 40126 Bologna, Italy; 2Department of Oncology, San Luigi Gonzaga Hospital, 10043 Orbassano, Italy; 3National Center for Drug Research and Evaluation, National Institute of Health (ISS), 00161 Rome, Italy

**Keywords:** chemotherapy, well differentiated neuroendocrine neoplasms, neuroendocrine tumors, treatment

## Abstract

Neuroendocrine tumors (NETs) are rare neoplasms with a wide spectrum of clinical behavior, from the long survival of well-differentiated NETs to the dismal prognosis of high-grade neuroendocrine carcinomas (NECs), being G3 NETs a recently recognized intermediate entity. While the role of chemotherapy is well established in NECs, data on NETs mostly derives from small studies, experts’ opinions, and extrapolating results from small-cell lung cancer studies. This narrative review aims to summarize available evidence about the use of chemotherapy in the setting of G1-2 NETs and G3 NETs. We performed literature research in PubMed Library for all articles published up to September 2022 about the efficacy of chemotherapy in NETs. Treatment regimens with STZ-5FU, CAPTEM, and anti-metabolite-based treatment are the most active and tolerated in gastroenteropancreatic NETs (GEP-NETs) G1-G2, while platinum-based regimens (FOLFOX/XELOX) and TEM/CAPTEM showed the best activity in thoracic NETs. Solid evidence about chemotherapy efficacy in G3 NETs is still lacking. Literature data support the use of chemotherapy in low-intermediate grade NETs after the failure of other therapies or if tumor shrinkage is needed. Studies assessing G3 NETs independently from NECs are needed to better understand the role of chemotherapy in this setting.

## 1. Introduction

Neuroendocrine neoplasms (NEN) comprise a heterogeneous family of neoplasms in terms of morphology, origin, clinical presentation, and response to treatments [1]. The most notable categories are well-differentiated NENs, or neuroendocrine tumors (NETs), and poorly differentiated ones, or neuroendocrine carcinomas (NECs) based on morphology. Overall, NETs share a well-differentiated morphology and a relatively good prognosis, if compared with NECs. Approximately 5–10% of NETs arise in the context of heritable tumor syndromes, such as multiple endocrine neoplasia (MEN) type 1, MEN-2, MEN-4, Von Hippel Lindau disease (VHL), and neurofibromatosis type 1 (NF1). Patients affected by these syndromes usually develop NET at a younger age as compared to those with sporadic NETs [2]. Although NETs are considered rare tumors, their incidence has substantially risen over the last decades (up to 6.98 new cases/100.000 inh/year), mainly due to the increased diagnosis rate of early-stage tumors [3,4,5]. Furthermore, the prevalence of NETs has also significantly increased, thanks to both the higher incidence and the improvement in the treatment armamentarium that prolonged the survival of patients with NET [3].

In addition to morphology, within the NET category, different biological behaviors can be identified according to the organ of origin and proliferation index. NETs of different origins (most commonly pancreas, GI tract, or lung) have different prognoses but also different susceptibility to treatments. Furthermore, since 2017, a new category of NETs characterized by well-differentiated morphology and Ki67 > 20% has been defined, G3 neuroendocrine neoplasms (NET G3). G3 NETs are high-grade tumors but remain distinct from NECs, as preliminary data suggest different genotypes, responses to treatment, and survival [4,5]. For instance, in a large retrospective series, median overall survival (mOS) for all patients was 23 months, significantly higher in NET G3 than NEC (99 vs. 17 months; *p* < 0.001) [6]. G3 NETs were first recognized by the 2019 WHO classification of tumors of the digestive system, they most commonly involve the pancreas, but they can occur throughout the GI tract [5].

Because G1-2 NETs have dramatically different clinical behavior as compared to G3 NETs, treatment approaches are different between these two entities. Somatostatin analogues (SSAs) are the mainstay of treatment for G1-G2 NETs thanks to the established antiproliferative activity, the favorable safety profile, and the inhibitory activity on hormone-producing tumors [7,8]. A higher incidence of biliary stone disease and related complications have been observed in patients with GEP-NETs treated with SSA [9,10]. Further treatment options include everolimus (an mTOR inhibitor), sunitinib (a multi-kinase inhibitor with antiangiogenic activity), and peptide receptor radionuclide therapy (PRRT) [11,12,13,14], with the latter favored over the former [15,16]. While chemotherapy is used in selected cases of G1-G2 NETs, it is the first-choice treatment for G3 NETs because of their aggressive behavior and high proliferation rate [6].

Several cytotoxic drugs have been proved to be active and are used for the treatment of G1-2 and G3 NETs. These include alkylating agents, such as streptozotocin (STZ), temozolomide (TEM), and dacarbazine (DTIC), antimetabolites, such as 5-fluorouracil (5-FU) and capecitabine (CAP), topoisomerase inhibitors, including etoposide and irinotecan, and platinum derivatives (mainly oxaliplatin, but also carboplatin and cisplatin) [17]. In this review, we sought to summarize available evidence about the use of chemotherapy in well-differentiated NETs and to provide guidance on its use in the setting of both G1-2 NETs and G3 NETs.

## 2. Materials and Methods

This narrative review was performed for all available prospective and retrospective studies, case reports, and review articles published up to September 2022 in PubMed. Data were extracted from the text and from the tables of the manuscript. The keyword search used included “chemotherapy”, plus: “neuroendocrine tumors”, “neuroendocrine neoplasm”, “carcinoid”, “neuroendocrine tumors and chemotherapy”, “neuroendocrine tumors and streptozotocin”, “neuroendocrine tumors and irinotecan”, “neuroendocrine tumors and platinum-based chemotherapy”, “neuroendocrine tumors and temozolomide”, “neuroendocrine tumors and capecitabine”, “neuroendocrine tumors and CAPTEM”.

## 3. Results

### 3.1. Role of Chemotherapy in Well-Differentiated G1 and G2 NETs

#### 3.1.1. Current Evidence in Well-Differentiated Pancreatic NETs

STZ-Based Regimens

Chemotherapy in patients with advanced pancreatic NET (panNET) was first investigated in 1980 in a phase II trial that compared STZ alone or in combination with 5-FU in 84 patients (Table 1 and Table 2). A higher objective response rate (ORR, 63% vs. 36%) and numerically longer overall survival (OS, median 26.0 vs. 16.5 months) were observed in the STZ-5-FU arm compared to the STZ arm, respectively [18]. A subsequent phase III trial showed the superiority of the combination of STZ and doxorubicin (DOXO) compared to STZ-5-FU in terms of ORR (69% vs. 45%), time to tumor progression (22 vs. 13 months), and OS (26 vs. 17 months) [19]. However, the use of STZ-DOXO is limited by the toxicity of this combination, e.g., severe nausea and vomiting.

A recent large retrospective study investigated outcomes of the combination 5- FU, DOXO, and STZ (FAS regimen) in 243 patients with well-differentiated panNETs and reported a median progression-free survival (mPFS) of 20 months and an mOS of 63 months [20]. Despite the fact that use of FAS as the first line of treatment was unsurprisingly associated with better PFS and OS when compared to later lines, no data was available about grading or Ki67 to explore the potential presence of a subgroup of patients who could benefit more from this intense chemotherapy regimen. Notably, the main registered severe adverse events were neutropenia (10%), nausea, and vomiting (5.5%), less than historically reported with the two-drug combinations. On the other hand, the addition of cisplatin (CDDP) to STZ-CAP did not demonstrate greater efficacy compared to STZ-CAP in a randomized study that included 86 patients with advanced NETs of pancreatic (N  =  41), gastroduodenal (N  =  17) or unknown primary (N  =  18), while the 3-drug regimen was significantly more toxic [29].

**Table 2 jcm-12-00717-t002:** Studies and trials of chemotherapy in G1-G2 NETs of both pancreatic and extrapancreatic origin.

Study	Type of Study	Study Population	Study Arms	Outcome		
				ORR	mPFS	OS
Bajetta et al. [30]	Phase II Prospective	Advanced neuroendocrine tumors (low and high-grade malignancy)	XELOX Low-grade	27%	NA	40 months
Ducreux et al. [31]	Phase II Prospective	well-differentiated endocrine carcinomas	LV5FU2 + irinotecan	NA	5 months	15 months
Kunz et al. [32]	Phase II Prosepctive	Advanced neuroendocrine tumors	Bevacizumab + FOLFOX	41.7% for pancreatic NET 13.6% for carcinoid	21 months for pancreatic NET 19.3 months for carcinoid	NA
E1281 Sun et al. [33]	Phase II-III	Advanced carcinoid tumors	STZ + 5FU vs. DOXO + FU	16 vs. 15.9%	5.3 vs. 4.5 months	24.3 vs. 15.7 months
FNCLCC–FFCD 9710 Dahan et al. [34]	Phase III	Metastatic carcinoid tumors	STZ-5FU vs. IFNα-2A	NA	5.5 vs. 14.1 months	30.4 vs. 44 months

panNETs = pancreatic neuroendocrine tumors; FOLFOX = folinic acid + fluorouracil + oxaliplatin; CAPTEM = capecitabine + temozolomide; ORR = objective response rate; GEP- NENs = gastroenteropancreatic neuroendocrine neoplasms; NEC = neuroendocrine carcinoma; DCR = disease control rate; PFS = progression-free survival; OS = overall survival; TVRR = tumor volume reduction rate; AUC = area under the ROC curve; UK = unknown origin.

NETs present brisk angiogenesis which can be effectively targeted by sunitinib [14]. The combination of chemotherapy with anti-angiogenic agents, such as sunitinib or bevacizumab, a humanized monoclonal antibody against the vascular endothelial growth factor (VEGF), for the treatment of patients with NET has been investigated. In the phase II BETTER trial, the combination of STZ-5-FU and bevacizumab was evaluated in 34 patients with panNET: after a 24-month follow-up, the median progression-free survival (PFS) was 23.7 months (the primary endpoint), ORR was 56% (DCR 100%), and OS rate at 2 years was 88%, without unexpected toxicity [21]. Although the addition of bevacizumab might be interesting to explore further, all taken together this data suggests that STZ-5FU is the best STZ-based combination so far in terms of activity-safety balance.

Regarding sequencing treatments, patients in the aforementioned retrospective series who received everolimus-based therapies after progressing on FAS (N = 108) had a mPFS of 10 months, which was similar to what was reported in prospective clinical trials of everolimus [12,13,20], while those who received TEM-based therapies after progressing on FAS (N = 60) achieved an ORR in 13% of cases and had an mPFS of 5.2 months, less than expected based on available data (see below). Nevertheless, 53% of patients received TEM-based treatment as fourth- or later treatment line, limiting the interpretation of these findings. To tackle the sequencing issue, the randomized phase III SEQTOR study (GETNE 1206) has recently compared everolimus followed by STZ-5FU chemotherapy upon progression (arm A, N = 71) with the reverse sequence (arm B, N = 70) in patients with panNET [22]. The trial, amended to assess PFS to the first line due to slow accrual, enrolled 20 patients with G1 panNET, 113 with G2 panNET, while in 8 patients, the tumor grade was unknown. Both sequences evidenced similar efficacy and no difference in 12-month PFS to the first line was observed (69% vs. 64%, respectively). Nevertheless, the ORR was higher in the STZ-5FU arm than in the everolimus arm (30% vs. 11%, respectively), suggesting that STZ-5FU may be preferred over everolimus in patients in whom there is a need for tumor shrinkage, e.g., to relieve symptoms.

TEM and CAPTEM

TEM is an orally available alkylating derivate of DTIC. It is used for the treatment of gliomas [35], and there is increasing evidence of its efficacy in the treatment scenario of NET, either alone or in combination with other agents, such as CAP (CAPTEM). TEM exerts its action by methylating the N-7 or O-6 guanine, causing damage to DNA that is repaired by the suicide enzyme O-6-methylguanine-DNA methyltranferase (MGMT), whose transcription is regulated through epigenetic silencing [36,37,38]. The MGMT expression loss or MGMT promoter methylation may predict TEM activity in NETs, despite the data being controversial, likely because of the different techniques used to assess MGMT status [39,40,41,42,43,44].

Diverse targeted agents such as thalidomide, bevacizumab, or everolimus, have been combined with TEM, with no clear benefit of these combinations over single-agent TEM [45,46,47,48]. On the other hand, the CAPTEM combination, with a reported ORR of 33–70%, median PFS of 18–20 months, and an mOS of 25.3–75.2 months, yielded the most interesting results [49,50,51,52,53]. The largest retrospective data comes from a study that included 462 patients with advanced NET (71% panNET) who received CAPTEM from 2008 to June 2019. In the overall cohort, complete responses were observed in 8 patients, partial responses in 204, and stable disease in 161, for an ORR of 51.5% with a median PFS and OS of 18 and 51 months respectively [23]. In addition, when compared to patients with extra-pancreatic NET (EP-NET), patients with a panNET had longer PFS (23 vs. 10 months, respectively) and OS (62 vs. 28 months, respectively) on CAPTEM. Recently, the results of the final analysis from the ECOG-ACRIN E2211 study, a randomized phase II trial of TEM and CAPTEM in patients with advanced G1-2 panNET, have been reported, providing the first prospective data in this setting [24]. Prior everolimus or sunitinib, no prior chemotherapy, and concurrent octreotide were allowed on trial, whose primary endpoint was PFS. A total of 144 patients (~50% G2, >90% with liver metastases) were randomized to receive TEM (N = 72) or CAPTEM (N = 72). Although there were some imbalances between the two arms, the trial met its primary endpoint of PFS which was longer in the CAPTEM arm compared to the TEM one (22.7 vs. 14.4 months, respectively), while a similar OS was observed in the two arms (58.7 vs. 53.8 months, respectively; HR: 0.58). Grade 3–4 adverse events (AEs) were more frequently reported in the CAPTEM arm than in the TEM one, as expected. Interestingly, MGMT deficiency, assessed with either immunohistochemistry or by promoter methylation, was associated with the increased response with both treatments.

Alternative schedules of TEM, such as the metronomic one, have been evaluated in a series of G2 and G3 NETs of mainly pancreatic and lung origin, and found to be active, tolerable, and to lead to clinical improvement as measured by ECOG PS, making it a suitable choice for patients with poor PS [54].

TEM and CAPTEM can also be combined with PRRT to improve activity and cover the latency of PRRT effect onset [55]. Recently, the Australasian Gastrointestinal Trials Group (AGITG) CONTROL NET study, a randomized (2:1) phase II open-label trial of 177Lu-Octreotate PRRT in combination with CAPTEM, enrolled 28 patients with panNET: 19 were randomized to receive CAPTEM + PRRT, while 9 received CAPTEM alone [25]. At a median follow-up of 57.5 months, PFS rate at 27 months was higher in the CAPTEM + PRRT than in the CAPTEM arm (61.1% vs. 33.3%, respectively; HR: 0.41 [95% CI: 0.15—1.12]). In addition, ORR at 12 months was also higher in the combination arm compared to the control arm (72.2% vs. 33.3%, respectively), while no grade 3–4 late hematologic AE was reported, despite this information should be taken with caution given the small sample size. However, the activity of CAPTEM + PRRT in panNETs might deserve to be tested in larger prospective cohorts. Overall, CAPTEM is an active treatment in panNETs, especially G2 ones or those with MGMT deficiency.

DTIC and other alkylating agents

In a phase II trial, 50 patients with advanced symptomatic or rapidly progressing functioning or non-functioning panNET have been treated with the synthetic alkylating agent DTIC at 850 mg/m^2^ dose [26]. Despite an ORR of 33% and a mOS of 19.3 months being observed, the unfavorable toxicity profile and the lack of data about the differentiation and grade of the tumors of the patients enrolled in the trial prevent DTIC routine use for the treatment of panNET, at least at the dose studied in this trial.

Recently, the combination of sunitinib with evofosfamide (TH-302), a DNA alkylator activated under hypoxic condition that are usually found in the tumor microenvironment, has been evaluated in the SUNEVO trial, which enrolled 17 treatment-naïve patients with pan-NET [27]. Despite meeting the first stage efficacy threshold (ORR 18%), the study was stopped early due to unacceptable toxicity (65% grade 3–4 AEs).

Anti-metabolite-based treatment

Anti-metabolites, such as 5-FU or CAP are often used in tumors of the gastrointestinal tract in combination with other drugs, such as oxaliplatin and irinotecan, and are also used in the treatment of panNET. Oxaliplatin-based combinations with 5FU (FOLFOX), gemcitabine (GEMOX), or CAP (XELOX/CAPOX), have shown various degrees of activity in NETs, including panNETs [30,32,56,57]. In a large retrospective series of 78 patients with mainly G2 NET who received oxaliplatin-based chemotherapy, an ORR of 26%, a median PFS of 8 months, and a mOS of 32 months were observed [58]. In the subgroup of patients with panNET (N = 46), the ORR was 22%. A phase II trial of XELOX enrolled 40 patients with NET, including 11 well-differentiated panNETs in which an ORR of 27% (three partial responses and five stable diseases) was observed [30].

A combined analysis of two phase II trials of bevacizumab combined with either FOLFOX (N = 36) or CAPOX (N = 40) in advanced NETs reported an ORR of 41% (N = 4/11) and a median PFS of 21 months in patients with panNET who received FOLFOX+ bevacizumab [32]. Overall, DCR was 94% with FOLFOX-bevacizumab and 78% with CAPOX-bevacizumab. Although these studies did not meet the primary endpoint, some patients might benefit from the addition of antiangiogenic therapy to chemotherapy, despite no predictive biomarker having been been identified.

Lastly, 5-FU and irinotecan (FOLFIRI regimen) have been investigated in two prospective trials which enrolled 20 patients with well-differentiated NET, including panNET [28,31]. The median PFS was 5.0–9.1 months, while DCR was 80%, with few observed objective responses (1 in each study), and an mOS of 15 months. Nevertheless, grade 3–4 hematological and non-hematological (mainly gastrointestinal) AEs were reported in 25–55% and 40–65% of patients, respectively. Due to its activity and toxicity profile, FOLFIRI is considered a treatment option for the failure of other therapies, while oxaliplatin-based chemotherapy is more widely adopted.

#### 3.1.2. Current Evidence in Well-Differentiated EP-NETs

The role of chemotherapy in the treatment of well-differentiated EP-NETs, is different whether we consider lung NETs (namely, typical [TC] and atypical carcinoids [AC]) or small intestine NETs (siNETs) (Table 2 and Table 3).

STZ-based regimens

A trial of STZ-5FU vs. STZ and cyclophosphamide enrolled 40 patients with siNET, 18 with NET of unknown origin, 17 with lung NET, and 10 with NET from other gastrointestinal sites [59]. ORR was 33% (N = 14/42) and 26% (N = 12/47) in patients treated with the STZ-5FU or STZ-cyclophosphamide, respectively. The ORR was higher with both regimens in the subgroup of patients with siNET (44% and 37%) than in patients with lung or unknown origin (both 29% and 0%). At crossover, more responses were observed in patients who switched to single-agent 5FU than in those who switched to single-agent cyclophosphamide (N = 2/11 vs. N = 0/8, respectively). No significant difference in survival was observed. The phase II-III EST 5275 study compared STZ-5FU with DOXO in 172 patients with metastatic carcinoid tumors (N = 61 small bowel, N = 18 other GI site, N = 18 lung, N = 75 unknown primary) [60]. No significant difference in OS (16 vs. 12 months) or in ORR (22% vs. 21%) was observed between the two arms of the study, with the two treatment options resulting in similar efficacy.

The phase II-III E1281 trial by Eastern Cooperative Oncology Group (ECOG) randomized 176 patients with advanced carcinoid tumors to receive STZ-5FU or DOXO-5FU chemotherapy [33]. Among the enrolled patients, 43 had a siNET, 22 a lung carcinoid, 12 a colorectal NET, 8 a pancreatic NET, and 78 a NET of other or unknown primary site. Despite no significant difference being observed in ORR (16.0% vs. 15.9%, respectively) or PFS (5.3 vs. 4.5 months, respectively) between the STZ-5FU and STZ-DOXO arms, OS was longer in the STZ-5FU arm compared to the STZ-DOXO one (24.3 vs. 15.7 months, respectively), raising the concern about the long-term toxicity of the latter combination. Moreover, the phase III randomized FNCLCC–FFCD 9710 trial of STZ-5FU vs. IFNα-2A enrolled 64 patients with metastatic carcinoids, 6 of which with panNET and 58 of which with EP-NET. PFS, the primary endpoint, was numerically shorter in the STZ-5FU arm when compared to the IFNα-2A arm (5.5 vs. 14.1 months), but did not reach statistical significance [34]. Given the different classifications used when these studies were carried out, it is difficult to assess the efficacy of STZ-based chemotherapy in patients G1-2 EP-NET; nevertheless, STZ-5FU might be active in this setting. Additionally, tumor heterogeneity prevents making definitive conclusions. Indeed, a retrospective study evaluated the efficacy of STZ-5-FU in combination with CDDP in 79 patients, 33 of whom had EP-NETs. ORR was 38% and 25% in panNET and non-pancreatic primary sites, respectively [65]. However, this study included both NETs and NECs, and ORR by subgroup was not reported. As previously mentioned, the addition of CDDP to STZ-CAP was not superior to STZ-CAP, while the toxicity of the three drug combination was consistently worse [29].

TEM and CAPTEM

The role of CAP and TEM has been investigated in carcinoids following trials that have showed the efficacy of these agents in panNETs. In a phase II trial of CAP as a single agent in 19 patients with EP-NET (N = 12 siNET), no objective response was observed and a DCR of 68% was observed [66], while CAP in association with bevacizumab in the BETTER trial slightly improved these outcomes to up to 18% ORR and 88% DCR among 49 patients with NET of gastrointestinal origin (N = 40 siNETs) [61]. The association of CAP at the metronomic schedule with bevacizumab and the SSA octreotide was investigated in the phase II XELBEVOCT prospective trial, demonstrating a similar ORR of only 18% [67].

In a retrospective series of 13 patients with lung NETs, an ORR of 31% has been reported with single-agent TEM, but prospective studies did not entirely confirm this result [68]. Despite combinations of TEM having been investigated in patients with NETs, few data about those with EP-NET are available [45,46]. More recently, in the ATLANT study, a single-arm, phase II trial of TEM and the SSA lanreotide in 40 patients with lung or thymus NETs (20% TC, 53% AC), DCR at 9 months, the primary endpoint, of 35% and a median PFS of 37.1 weeks have been reported [62]. The trial met its non-acceptability threshold, but did not reach the clinical significance threshold in the primary analysis. Nevertheless, a sensitivity analysis extended the DCR in the period between 7.5 and 10.5 months reaching 45% and the clinical significance threshold, suggesting that the combination of TEM and lanreotide might be active in this setting. Notably, the treatment was tolerable, with mainly grade 1–2 reported AEs.

Because of the disappointing results of CAP and TEM as monotherapy, outcomes of CAPTEM were investigated in a retrospective series of 65 patients with NET, 19 of whom had EP-NET [69]. In this study, the ORR was lower in the EP-NET group than in the panNET group (37% vs. 48%, respectively). Furthermore, a small retrospective study that investigated CAPTEM in advanced lung NETs reported an ORR of 18%, an mPFS of 9.0 months, and a mOS of 30.4 months [63]. In addition, this combination has been evaluated in a phase II trial in a patient population of eleven panNETs, 12 EP-NETs, two thyroid carcinomas, and three pituitary adenomas. ORR was 33% and 36% in lung NETs and panNETs, respectively [49]. Given these findings, the role of CAPTEM in EP-NET is still unclear, though it may be an option for patients with lung NETs.

A phase I-II trial of CAPTEM associated with PRRT with 177-Lu-Octreotate enrolled 35 patients with advanced NET [70]. The ORR was 53%, including 15% complete responses, which was higher in patients with gastric or panNET (82%) as compared to those with siNET (26%) or lung NET (0%, the best response was stable disease in the two lung patients included). Furthermore, the AGITG CONTROL NET study randomized (2:1) patients affected by midgut NETs (N  =  45) to receive PRRT + CAPTEM vs. PRRT alone in phase II randomized non-comparative trial [25]. ORR with the PRRT + CAPTEM combination was higher than with PRRT alone (34% vs. 23%); however, that did not translate into a PFS benefit (62% vs. 60% at 36 months). However, bone marrow toxicity was concerning in the combination arm, with an incidence of myelodysplastic syndrome as high as 10%.

DTIC

Data about the use of DTIC in EP-NET come solely from a phase II trial evaluating the response rate and toxicity of high and low doses of DTIC in 56 patients with advanced carcinoids with different primary sites. The ORR was 20% for high-dose DTIC and 16% for low-dose DTIC, with a median overall survival of 20 months [64]. The most common adverse events were nausea and vomiting (88% of patients). Therefore, the benefits of treatment with DTIC are marginal, apparently not balanced by toxicity.

Platinum-based regimens

Platinum-etoposide combination (PE) has been investigated in small retrospective studies in TC and AC, with observed ORR ranging from 23 to 39% and a median PFS of 7 months [53,71,72,73]. A higher degree of activity of PE chemotherapy is observed in lung or thymus NETs, with an ORR of 39% in this group vs. 27% in the panNET group [57]. In a prospective trial of XELOX in patients with lung NET, ORR was 60% with a tolerable safety profile [30], suggesting that FOLFOX might be a more active and tolerable treatment in patients with lung NET when compared to PE, taking into account the limitations of cross-study comparisons.

#### 3.1.3. Chemotherapy in G1 and G2 NETs: Ongoing Trials

Diverse ongoing trials of chemotherapy, mostly in combination with other agents, are enrolling patients with G1-2 NETs (Table 4). With respect to alkylating agents, the SONNET study is an ongoing phase II single-arm trial of the SSA lanreotide and TEM in 57 patients with progressive G1-2 GEP-NET, whose primary endpoint is DCR at 6 months (NCT02231762). Additionally, the COMPOSE study aims to assess efficacy, safety, and outcomes of PRRT with 177Lu-Edotreotide used as first- or second-line treatment compared to CAPTEM, EVE, and FOLFOX (control arm) in patients with well-differentiated but aggressive G2 and G3, (SSTR+), and GEP-NETs (NCT04919226).

In the anti-metabolite scenario, the BETTER-2 trial randomized patients with well-differentiated G1-3 panNET to receive chemotherapy regimens like STZ–5-FU vs. CAPTEM, with or without bevacizumab, in a two-by-two design; the primary endpoint is PFS (NCT03351296). TAS 102, an anti-metabolite combination of Trifluridine/Tipiracil, which is currently approved for the treatment of advanced colorectal adenocarcinoma, is also under investigation for the treatment of patients with NET [74]. The combination of TAS 102 and TEM has been evaluated in a phase I trial dose escalation that enrolled 13 NETs patients [75]. TEM/TAS 102 were well tolerated, and preliminary activity data have shown 1 PR (8%) and a DCR of 92% (N = 12/13) in this small cohort. The enrollment for the expansion of the G1-G2 NET cohort is ongoing.

To prospectively clarify the role of MGMT status as a predictive factor, the MGMT NET trial aims to explore the contribution of MGMT gene methylation on tumor tissue in predicting the objective response in patients treated with alkylating agents and to compare treatment with alkylating agents to oxaliplatin in patients with a duodeno-pancreatic, lung or unknown primitive NET (NCT03217097).

Of interest, the CapTemY90 trial will assess the efficacy of trans-arterial radioembolization (TARE) during CAPTEM treatment in patients with liver-dominant G2 NET metastases from any primary. The assessment of TARE suitability has to be performed during the first cycle of systemic therapy. Subsequently, based on tolerance to the treatment and persistence of eligibility, TARE will be performed on Day 7 of Cycle 2, with an additional treatment on Day 7 of cycle 3 or 4 if needed to treat the entire tumor burden (NCT04339036).

#### 3.1.4. Role of Chemotherapy in Neoadjuvant/Adjuvant Setting in G1 and G2 NET

Adjuvant therapy is a regimen given after the main cancer treatment with the aim to increase the chance of curing or delaying disease recurrence, killing circulating tumor cells and micrometastases. After complete resection of a GEP-NET, the risk of recurrence is 57% for panNETs and 53% for siNETs [76]. The most relevant risk factors for recurrence are grade, vascular invasion, perineural invasion, and stage [77]. Moreover, after resection of liver metastases, the median time to recurrence is 15.2 months (95% CI: 11.2–19.2 months), and 5- and 10-year overall recurrence rate is 94 and 99%, respectively [78,79].

A retrospective series reported on the role of adjuvant therapy with STZ/5-FU in patients with panNETs and siNETs after resection of liver metastases and showed that relapse-free survival (RFS) at 5 years was similar between the adjuvant treatment group (N = 29, 20%) and the observation group (N = 23, 38%) [80]. Nevertheless, these results should be interpreted with caution: indeed, performing an adjuvant prospective trial in patients with NET has several shortcomings from the methodological standpoint, because of the rarity of the disease and the long relapse-free survival. On the other hand, retrospective series are difficult to interpret because of the heterogenous practice of different centers, and the bias of treatment allocation that can only be partially controlled with statistical corrections [81]. In summary, there are no randomized prospective trials investigating adjuvant therapy in patients with G1-2 NET, and the available data pooled together different NET types and have conflicting results. Thus adjuvant chemotherapy is not recommended by either ENETS or NCCN guidelines [6,82].

On the other hand, neoadjuvant treatment is generally provided in patients with tumors before surgery to decrease tumor size and facilitate the surgical procedure. As per the adjuvant setting, definitive evidence in NETs is still lacking and neoadjuvant therapy is not routinely recommended [83]. In a retrospective study of twenty-nine patients with locoregionally advanced panNET who had received chemotherapy with 5FU-DOXO-STZ (FAS) in preoperative setting, only two patients achieved a radiological partial response according to RECIST criteria and six had an improvement of the tumor-vascular interface, while stable disease was observed in 90% of patients [84]. Of the fourteen patients who underwent surgery, a complete resection (R0) was achieved in only nine patients. In conclusion, patients receiving FAS appeared to rarely progress during chemotherapy, but considerable tumor shrinking was uncommon. Preoperative administration of FAS has been evaluated also in patients undergoing liver surgery for advanced synchronous liver metastases from panNET, resulting in 17/27 partial remission, with a significant improvement in OS and RFS [85]. Another retrospective study evaluated different neoadjuvant chemotherapy regimens (i.e., 5-FU, STZ, DOXO, cisplatin, etoposide, and oxaliplatin) in 42 patients with G1/G2 locally advanced panNET with segmental portal hypertension [86]. R0 resection was achieved in 13 out of 28 cases who underwent surgery, but this did not translate into an improvement in OS.

Available evidence does not suggest the use of neoadjuvant treatment in patients with G1-G2 NET, irrespective of the primary site [87]. This strategy might be, however, pursued in selected cases of patients with panNET in which tumor shrinkage might help achieve a radical resection, as surgery is the only chance of cure for these patients. Nevertheless, there is no data to inform treatment choice in this setting, thus these cases should be discussed on single-case bases in multidisciplinary tumor boards of NET-expert Centers.

In lung NETs, many large retrospective studies reported no benefit of adjuvant treatments in both TCs and ACs [88,89,90,91]. As a consequence, authors do not recommend routine adjuvant therapy in lung NETs, which may be considered only in selected fit patients with a particularly high risk of relapse (e.g., AC with N2 involvement) after multidisciplinary discussion [92].

### 3.2. Role of Chemotherapy in Well-Differentiated G3 NETs

#### 3.2.1. Current Evidence

Clinical management of NET G3 is challenging due to the lack of data and the fact that treatment is still not standardized.

Chemotherapy in early-stage disease

The recommended treatment for localized NEN, according to European and American guidelines, is surgical resection, regardless of the tumor grading [93,94]. When a tumor shrinkage is needed before the surgery, chemotherapy is usually considered in G3 NETs. In fact, neoadjuvant treatment could lead to a reduction of the tumor mass, allowing a secondary surgery [95]. In addition, according to the European Neuroendocrine Tumor Society (ENETS) Guidelines, a systemic cytotoxic treatment is recommended in progressive or bulky advanced NETs localized in the pancreas or in other sites when there is a high Ki-67, rapidly progressive disease, and after the failure of other therapies if SSTR-imaging is negative [96].

Platinum-based chemotherapy

In advanced disease, a first-line chemotherapy regimen with PE represents the standard in neuroendocrine carcinomas. However, EP seems to be less effective in well-differentiated G3 NETs than poorly differentiated NECs according to real-world studies that reported an objective response rate (ORR) of up to 44% for highly proliferative NECs vs. 24% in NET G3 (Table 5) [96]. Notably, data from a retrospective Asiatic database including 100 patients showed that G3 panNETs with Rb loss and those with mutated KRAS showed significantly higher ORR to PE than those without (Rb loss, 80% vs. normal Rb, 24%; mutated *KRAS*, 77% versus wild type, 23%), suggesting a predictive role of these biomarkers that could help clinicians in treatment choice [97].

The rationale of using other platinum-based regimens (consisting of oxaliplatin and a fluoropyrimidine) in this setting is based on favorable activity data in well-differentiated high-grade NETs, both in first and subsequent lines of therapy, and the better tolerability as compared to EP [58,98]. For instance, a retrospective analysis of 72 advanced GEP-NETs patients, of whom 55.5% with a G3 NET, treated with FOLFOX, mainly as upfront treatment, reported a disease control rate (DCR) of 75% [99].

Antiangiogenic therapy

The addition of the antiangiogenic bevacizumab to CAPOX/FOLFOX was tested in two phase II trials showing favorable ORRs (19% for CAPOX and 50% for FOLFOX) and prolonged DCR (respectively 78% and 98%) in previously progressing well-differentiated panNET patients. G3 panNETs were not included in the studies, but a randomized trial is going to assess the efficacy of the antiangiogenic in addition to chemotherapy in this setting (*see “Ongoing trials on chemotherapy in G3 NETs”).*

Irinotecan

Several retrospective studies, mostly from Asia, have assessed the efficacy of first-line irinotecan and platinum (IP) in NECs with promising results. For instance, a phase II study of irinotecan in combination with cisplatin in high-grade NETs, that included 20 patients with extrapulmonary NECs, reported an ORR of 58% [103]. While a current randomized trial is exploring this combination in poorly differentiated NECs, the efficacy in G3 NETs remains uninvestigated [104]. In contrast, activity data of the FOLFIRI regimen are limited in well-differentiated low-intermediate grade NETs [28].

STZ

Historically, the combination of STZ with other cytotoxic drugs, mainly 5-fluorouracil (5-FU) was used in patients with advanced panNET. The employment of this drug is occasionally limited due to renal toxicity, but data on efficacy are solid, with the combination STZ-5FU showing ORR of 63% and mOS of 26 months [18,19]. Recently, a Japanese retrospective study evaluated 20 patients with unresectable or metastatic pancreatic neuroendocrine neoplasms that had received weekly STZ and oral S-1 therapy (STS1). Even if only three cases of G3 NET were included in the analyses, the results showed that this combination is effective and safe, representing a potential new therapeutic option for patients with advanced P-NETs [100].

TMZ

The efficacy of TMZ has been investigated as a single agent or in combination with other drugs (CAP, bevacizumab, EVE, and thalidomide), primarily in panNETs, showing considerable efficacy, and in few small studies in EP-NETs, with less encouraging results [45,46,48,105]. The most solid evidence of the efficacy of CAPTEM in panNETs has been recently provided by the E2211 randomized phase II trial conducted by the ECOG-ACRIN Cancer Research Group (ORR 40%, mPFS 22.7 months, mOS 58.7 months), though G3 NETs were not included [24]. Even if based on a small cohort of 16 patients, retrospective data showed that treatment with alkylating agents, such as TMZ, seems to be more promising in G3 P-NETs as compared to NECs, with an ORR of 50% [101]. Alongside, real-world data from the Italian group on 100 patients treated with TMZ or CAPTEM, including nine G3 NETs, showed a great benefit using TMZ-based treatment in this subgroup, with a median overall survival of 35 months (four times higher than G3 NEC patients) [102].

#### 3.2.2. Chemotherapy in G3 NETs: Ongoing Trials

Being only recognized in 2019 [95], well-differentiated G3 NETs have been previously merged into the heterogeneous “NEC” category and clinical trial, specifically dedicated to this entity are still lacking. Remarkably, results of the planned interim analysis of the phase II randomized trial ECOG-ACRIN EA2142 (NCT02595424) were recently reported showing that CAPTEM does not appear to be superior to PE chemotherapy as front-line treatment for patients with advanced well-differentiated G3 GEP-NETs or poorly differentiated, non-small cell NECs (Table 3) [106]. However, these results do not allow for the comparison of the efficacy of CAPTEM vs. PE in the subgroup of well-differentiated G3 NETs. The BETTER-2 trial (NCT03351296) is an ongoing phase II trial that randomized well-differentiated G1-3 panNET patients to receive STZ–5-FU vs. CAPTEM, with or without bevacizumab, in a two-by-two design, being PFS the primary endpoint. A prospective phase I/II study will evaluate the safety and activity of lurbinectedin in combination with irinotecan in pretreated patients with selected advanced solid tumors, including G2-G3 NETs of GEP origin or unknown primary site (NCT02611024). A retrospective study is assessing the efficacy of chemotherapy on well-differentiated G3 GEP-NETs from the prospective data of the French neuroendocrine tumors registry (GTE) (NCT04365023).

Recently, the combination of nivolumab plus platinum-based chemotherapy showed encouraging activity and prolonged survival benefits in patients with advanced G3 NETs of the GEP tract or of unknown origin, including poorly differentiated NECs [107]. Randomized trials are warranted to confirm treatment efficacy and ongoing translational studies may confirm the benefit of this treatment strategy in well-differentiated G3 NETs.

Lastly, the COMPOSE study will determine if PRRT with 177Lu-Edotreotide is superior to the best standard of care (mainly chemotherapy) as a first or second line of treatment in patients with well-differentiated aggressive G2 and G3, somatostatin receptor-positive (SSTR+), GEP-NETs.

## 4. Conclusions

In G1-G2 NETs, chemotherapy is not the first choice of treatment but is reserved to select cases. Advanced panNETs can be preferentially used in G2 tumors, especially with high proliferation rates (Ki-67 > 10%), and if rapidly progressive or bulky, symptomatic tumors. TEM and CAPTEM are the most useful regimens in this setting.

Regarding EP-NETs, chemotherapy showed limited efficacy, but STZ-based regimens or TEM may be considered in advanced progressive disease after the failure of other therapeutic options. Ongoing trials are needed to clarify the role of chemotherapy in this context and identify efficacy markers in these poorly-understood tumors.

G3 NETs have been partially explored so far, hence treatment strategy is still not consensus-based and recommendations are exclusively based on experts’ opinions (Figure 1). However, this recently recognized entity may have a different response to chemotherapy and survival than NECs, and studies assessing G3 NETs independently are needed for a better understanding of the most appropriate treatment approach.

## Figures and Tables

**Figure 1 jcm-12-00717-f001:**
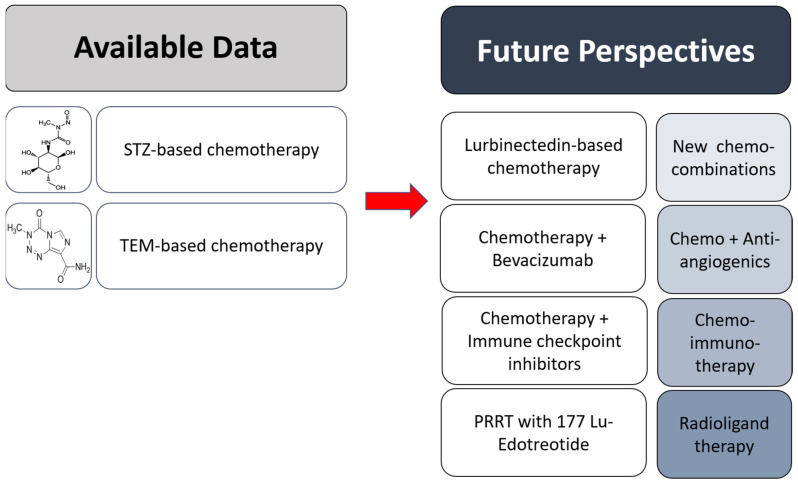
Current evidence and future perspectives for the medical treatment of advanced NET G3. Chemo: Chemotherapy, TEM: Temozolomide, PRRT: Peptide Receptor Radionuclide Therapy, STZ: Streptozocin.

**Table 1 jcm-12-00717-t001:** Studies and trials of chemotherapy in G1-G2 PanNETs.

Study	Type of Study	Study Arms	Outcome		
			ORR	mPFS	OS
Moertel et al. [18]	Phase II Prospective	STZ vs. STZ + 5FU	63 vs. 36%	NA	26 vs. 16.5 months
Moertel et al. [19]	Phase III Prospective	STZ + 5FU vs. STZ + DOXO	69 vs. 45%	20 vs. 6.9 months	26 vs. 17 months
Rogers et al. [20]	Retrospective	STZ + 5FU + DOXO (FAS regimen)	41%	20 months	63 months
BETTER Ducreux et al. [21]	Phase II Prospective	STZ + 5FU + Bevacizumab	56%	23.7 months	OS rate (2 years) 88%
SEQTOR Grupo Espanol de Tumores Neuroendocrinos, clinicaltrials.gov; NCT02246127 [22]	Phase III Prospective	Everolimus→STZ + 5FU (arm A) STZ + 5FU→Everolimus (arm B)	11 arm A vs. 30% arm B	mPFS rate 69 arm A vs. 64% arm B	NA
Al-Toubahet al. [23]	Retrospective	CAPTEM	51.5%	18 months	51 months
ECOG-ACRIN E2211 Kunz et al. [24]	Phase II Prospective	CAPTEM vs. TEM	40% vs. 34%	PFS 22.7 vs. 14.4 months	53.8 vs. 58.7 months
AGITG CONTROL NET Pavlakis et al. [25]	Phase II Prosepctive	CAPTEM + PRRT vs. CAPTEM	72.2 vs. 33.3%	mPFS rate (27 months) 61.1 vs. 33.3%	HR 1.28 (27 months)
Study of the Eastern Cooperative Oncology Group-E6282 Ramanathan. [26]	Phase II Prospective	Dacarbazine	33%	10 months	19.3 months
SUNEVO Grande et al. [27]	Phase II Prospective	Sunitinib + Evofosfamide (TH-302)	Stopped for toxicity		
Brixi-Benmansour et al. [28]	Phase II Prospective	FOLFIRI		9.1 months	65% months

panNETs = pancreatic neuroendocrine tumors; FOLFOX = folinic acid + fluorouracil + oxaliplatin; CAPTEM = capecitabine + temozolomide; ORR = objective response rate; GEP- NENs = gastroenteropancreatic neuroendocrine neoplasms; NEC = neuroendocrine carcinoma; DCR = disease control rate; PFS = progression-free survival; OS = overall survival; TVRR = tumor volume reduction rate; AUC = area under the ROC curve; UK = unknown origin.

**Table 3 jcm-12-00717-t003:** Studies and trials of chemotherapy in G1-G2 EP-NETs.

Study	Type of Study	Study Population	Study Arms	Outcome		
				ORR	mPFS	OS
Moertel et al. [59]		Carcinoid tumors	STZ + 5FU vs. STZ + cyclophosphamide	33 vs. 26%	NA	NA
EST 5275 Engstrom et al. [60]	II-III phase Prospective	Carcinoid tumor	STZ + 5FU vs. DOXO	22 vs. 21%	7.75 vs. 16 months	16 vs. 12 months or 6.5 vs. 12 months
BETTER Mitry et al. [61]	II phase Prospective	GI-NETs	CAP + Bevacizumab	NA	23.4 months	Survival rate (2 years) 85%. mOS not reached
ATLANT Ferolla et al. [62]	II phase Prospective Lung NETs	Lung NETs	TEM + SSA	NA	9.3 months	NA
Papaxoinis et al. [63]	Retrospective	Lung NETs	CAPTEM	18%	9.0 months	30.4 months
AGITG CONTROL NET Pavlakis et al. [25]	II phase Prospective	siNETs	PRRT + CAPTEM vs. PRRT	34 vs. 23%	mPFS rate (36 months) 62 vs. 60%	HR 0.61 (36 months)
Bukowski et al. [64]	Phase II Prospective	Metastatic carcinoid tumors	Dacarbazine	20% high doses 16% low doses	NA	20 months

panNETs = pancreatic neuroendocrine tumors; FOLFOX = folinic acid + fluorouracil + oxaliplatin; CAPTEM = capecitabine + temozolomide; ORR = objective response rate; GEP- NENs = gastroenteropancreatic neuroendocrine neoplasms; NEC = neuroendocrine carcinoma; DCR = disease control rate; PFS = progression-free survival; OS = overall survival; TVRR = tumor volume reduction rate; AUC = area under the ROC curve; UK = unknown origin.

**Table 4 jcm-12-00717-t004:** Ongoing studies on chemotherapy use in well-differentiated neuroendocrine tumors.

Study	Study Population	Study Arms	Primary End-Point
SONNET, phase II trial (NCT02231762)	Progressive G1-2 GEP-NETs	Lanreotide + TEM (single arm)	DCR at 6 months
COMPOSE, phase III trial (NCT04919226)	G2-G3 GEP-NETs (SSTR+)	^177^Lu-edo-PRRT vs. SOC (CAPTEM/EVE/FOLFOX)	PFS
BETTER 2, phase II trial (NCT03351296)	G1-3 panNETs	-CAPTEM +/− BEV -STZ-5FU +/− BEV	PFS
MGMT-NET, prospective interventional trial (NCT03217097)	G1-3 panNETs—EP-NETs	- *Unmethylated MGMT* Oxaliplatin-based chemotherapy vs. Alkylating-based chemotherapy (1:1) - *Methylated MGMT* Oxaliplatin-based chemotherapy vs. Alkylating-based chemotherapy (1:2)	OR in NETs treated with alkylating-based chemotherapy according to MGMT methylation status
CapTemY90, phase II trial (NCT04339036)	liver-dominant metastases from G2 p-EP-NETs	Oral CapTem + Y90 Radioembolization (single arm)	Intra-hepatic PFS
ECOG-ACRIN EA2142, phase II trial (NCT02595424)	G3 GEP-NETs and non-small cell NECs	PE vs. CAPTEM	PFS
NCT02611024, phase I/II trial	Pretreated advanced solid tumors (including NETs)	Lurbinectedin + irinotecan (single arm)	MTD
TNE-bien-DIF, retrospective study (NCT04365023)	G3 GEP-NETs	- first-line platinum-based chemotherapy - first-line non-platinum-based chemotherapy	OS

NETs = neuroendocrine tumors, NECs = neuroendocrine carcinomas, GEP = gastroenteropancreatic, SSTR = somatostatine receptor, panNET = pancreatic NET, EP-NET = extrapancreatic, FOLFOX = Folinic acid + Fluorouracil + Oxaliplatin, CAPTEM = Capecitabine and Temozolomide, PRRT = Peptide Receptor Radionuclide Therapy, EVE = everolimus, STZ = Streptozocin, 5FU = 5-fluorouracil, BEV = bevacizumab, MGMT = O^6^-Methylguanine-DNA Methyltransferase, DCR = disease control rate, OR = objective response, PFS = progression-free survival, MTD = maximum tolerated dose, OS = overall survival, PE = platinum-etoposide.

**Table 5 jcm-12-00717-t005:** Studies and trials of chemotherapy in the treatment of G3 NET.

Study	Type of Study	Study Population	Study Arms	Endpoints of the Study
Al-Toubah et al. [98]	Retrospective	G1, G2, G3 panNETs	FOLFOX	ORR
Merola et al. [99]	Retrospective	G1, G2, G3 GEP-NENs	FOLFOX	DCR, PFS, OS
Ono et al. [100]	Retrospective	G1, G2, G3 panNETs, G3 NEC	Weekly streptozotocin + oral S1	TVRR, PSF, OS
Raj et al. [101]	Retrospective	G3 panNETs, G3 NEC	Platinum-based regimens or alkylating agents (dacarbazine and temozolomide)	OS, ORR
Bongiovanni et al. [102]	Retrospective	G1, G2, G3 NENs, G3 NEC	Temozolomide alone or in combination with capecitabine	OS, PFS, ORR, DCR
NCT03980925, phase II	Prospective non-randomized	G3 GEP/UK-NENs	First line Nivolumab + Carboplatin AUC 5 + Etoposide followed by nivolumab maintenance (single arm)	OS, PFS, ORR

panNETs = pancreatic neuroendocrine tumors; FOLFOX = folinic acid + fluorouracil + oxaliplatin; CAPTEM = capecitabine + temozolomide; ORR = objective response rate; GEP- NENs = gastroenteropancreatic neuroendocrine neoplasms; NEC = neuroendocrine carcinoma; DCR = disease control rate; PFS = progression-free survival; OS = overall survival; TVRR = tumor volume reduction rate; AUC = area under the ROC curve; UK = unknown origin.

## Data Availability

Not applicable.

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
