# Peer review of "Chemotherapy in Well Differentiated Neuroendocrine Tumors (NET) G1, G2, and G3: A Narrative Review"

_jcm, 2023, doi:10.3390/jcm12020717_

Round 1

Reviewer 1 Report

Overall the authors wrote a good review on NETs 

While the article is well written and summarizes important concepts in their treatment of NET it is not very interesting in the way it is written. It lacks a good amount of tables and schemas which would make it interesting to the reader. Important treatments such as Lu 177 are not adequately described. Overall doesnt add much to existing reviews on the same topic. 

Author Response

We thank all the reviewers for their comments and suggestions. By addressing them, we feel that the manuscript has significantly improved and hope it is now suitable for publication in the Journal of Clinical Medicine.

Table 1, 2, 3 and 5 with the available evidence that support the use of chemotherapy in NET G1-2 and NET G3 have been added. Furthermore, we added Figure 1 that depicts currently used chemotherapy in NET G3 and potential future strategies, including 177Lu PRRT. As the focus of this review is on the use of chemotherapy in well-differentiated NEN, PRRT with 177Lu has been discussed only in combination with chemotherapy (lines 238-249, 387-396, 421-424), rather than as single agent or in combination with agents other than chemotherapy.

Reviewer 2 Report

This paper presents a very helpful overview, in the form of a narrative review, of the role of chemotherapy in the management of well differentiated neuroendocrine tumours (NETS). In particular it includes comment on the relatively recently described entity of well differentiated NET G3. 

As a general comment, the paper would benefit from a review of spelling and grammar, as well as restructuring of the paragraphs, some of which are very long. This would make it rather easier to read.

The abstract provides a concise overview of the aims of the paper and the main findings. In the background section, care is required with the nomenclature. My understanding is that the collective name for neuroendocrine tumours (NET) and neuroendocrine carcinomas (NEC) is neuroendocrine neoplasm (NEN). Neuroendocine tumours are by definition all well differentiated although now include G3 tumours, and it is this group that is clearly the focus of this article. Whilst reasonable to refer to NEC here, I think it is important to be clear that this group is distinct from NETs.

The introduction gives helpful background information regarding NETs, including a description of the relatively new entity of G3 NET and its prognostic significance. There is also a helpful overview of existing treatment options, beyond chemotherapy. It might be helpful to include here more reference to the anatomical classification- ie the distinction between pancreatic, GI and lung NETs- and the significance in terms of differential response to treatment. Admittedly this is covered elsewhere in the paper, but I think it would be helpful to acknowledge this in the introduction.

The methods section clearly describes the search strategy, and is clear that this is a narrative review.

The results are presented broken down by tumour grade (G1/2 vs G3) and anatomical site (pancreatic vs extra-pancreatic)  which appears reasonable., although it might have been helpful to further differentiate extra-pancreatic into GI and lung subgroups given significant differences. Helpful mention is also made of on-going clinical trials, along with helpful comment on the role of adjuvant/neoadjuvant chemotherapy for G1/2 tumours. 

The review appears to be comprehensive, with no glaring omissions, and includes both retrospective and prospective studies.

As alluded to above, the text is rather dense and would benefit from being broken up into shorter paragraphs. In addition it might be helpful to include more tables, summarising the studies in each section, in a similar format to table 1 which  summarises on-going trials.

The conclusions drawn at the end of the paper all appear to be entirely reasonable and justified by the evidence that has been reviewed.

Overall I think that this is a very helpful overview, but would benefit from some minor editing.

Author Response

We thank all the reviewers for their comments and suggestions. By addressing them, we feel that the manuscript has significantly improved and hope it is now suitable for publication in the Journal of Clinical Medicine.

This paper presents a very helpful overview, in the form of a narrative review, of the role of chemotherapy in the management of well differentiated neuroendocrine tumours (NETS). In particular it includes comment on the relatively recently described entity of well differentiated NET G3. As a general comment, the paper would benefit from a review of spelling and grammar, as well as restructuring of the paragraphs, some of which are very long. This would make it rather easier to read.

We thank the reviewer for this suggestion. We checked grammar and restructured paragraphs throughout the paper to make it easier to read.

The abstract provides a concise overview of the aims of the paper and the main findings. In the background section, care is required with the nomenclature. My understanding is that the collective name for neuroendocrine tumours (NET) and neuroendocrine carcinomas (NEC) is neuroendocrine neoplasm (NEN). Neuroendocine tumours are by definition all well differentiated although now include G3 tumours, and it is this group that is clearly the focus of this article. Whilst reasonable to refer to NEC here, I think it is important to be clear that this group is distinct from NETs.

We thank the reviewer for this comment. The NEN morphological classification into NET and NEC has been clarified, highlighting that NETs and NECs are different entities (line 31-35).

The introduction gives helpful background information regarding NETs, including a description of the relatively new entity of G3 NET and its prognostic significance. There is also a helpful overview of existing treatment options, beyond chemotherapy. It might be helpful to include here more reference to the anatomical classification- ie the distinction between pancreatic, GI and lung NETs- and the significance in terms of differential response to treatment. Admittedly this is covered elsewhere in the paper, but I think it would be helpful to acknowledge this in the introduction.

Thank you for this suggestion. A sentence about the differential sensitivity to treatment of NET fo different organ of origin has been added in the background (lines 50-52).

The methods section clearly describes the search strategy, and is clear that this is a narrative review. The results are presented broken down by tumour grade (G1/2 vs G3) and anatomical site (pancreatic vs extra-pancreatic)  which appears reasonable., although it might have been helpful to further differentiate extra-pancreatic into GI and lung subgroups given significant differences. Helpful mention is also made of on-going clinical trials, along with helpful comment on the role of adjuvant/neoadjuvant chemotherapy for G1/2 tumours. 

We thank the reviewer for this comment. Despite we acknowledge that further grouping GI NETs from lung NETs could have been helpful, many of the studies included in these paragraphs enrolled patients with NETs from different primary sites and studies should have been repeated for the great part in both sections for GI and lung NETs. However, in an attempt to keep GI and lung NETs as separated as possible, in each paragraph GI-only results are reported before lung-only results, where available.

The review appears to be comprehensive, with no glaring omissions, and includes both retrospective and prospective studies. As alluded to above, the text is rather dense and would benefit from being broken up into shorter paragraphs. In addition it might be helpful to include more tables, summarising the studies in each section, in a similar format to table 1 which  summarises on-going trials.

We thank the reviewer for these comments. As stated above, Table 1, 2, 3 and 5 with the available evidence that support the use of chemotherapy in NET G1-2 and NET G3 have been added. Furthermore, we added Figure 1 that depicts currently used chemotherapy in NET G3 and potential future strategies. Lastly, we’ve also broken up paragraphs into shorter ones, to make the paper easier to read.

The conclusions drawn at the end of the paper all appear to be entirely reasonable and justified by the evidence that has been reviewed. Overall I think that this is a very helpful overview, but would benefit from some minor editing.

Reviewer 3 Report

In their review article, the authors summarize current chemotherapeutic concepts for the treatment of neuroendocrine tumors (G1-G3). Although this is certainly a very exciting topic, numerous reviews exist on this subject. The authors also published a review on this very topic in 2021, which seems to me to be very redundant, especially in the sections on pancreatic and extra-pancreatic G1 and G2 NETs, so that 2/3 of the manuscript does not really delineate the already published content. Only the paper on the topic of NET-G3 distinguishes these two manuscripts. In addition, there are individual sentences that remain incomprehensible to the reader. As an example, lines 40-41 would be This sentence states that the increasing prevalence is due to the indolent nature of the tumors, but also due to the improvement of therapeutic strategies. This linkage or assertion is very strange in my eyes. The sentence in lines 36-37 is also incomprehensible or grammatically incorrect.
Line 1 should also use the correct definition for the basic name of all NETs and NECs, neuroendocrine neoplasms (NEN).

Author Response

We thank all the reviewers for their comments and suggestions. By addressing them, we feel that the manuscript has significantly improved and hope it is now suitable for publication in the Journal of Clinical Medicine.

 We agree that this is an exciting topic that deserved to be covered in order to provide an updated review of the available chemotherapy-based strategies in patients with NET. Despite a partial overlap with a previous work, the scenario of the available evidence in the G1-G2 NET setting evolves quickly with new data (e.g., Kunz et al JCO 2022) that we believe it deserves to be updated.

Also, the manuscript has been proofread to make sentences like those highlighted clearer. In particular, lines 36-37 and 40-41 now read as follows (lines 37-44 in the revised manuscript):

“Patients affected by these syndromes usually develop NET at a younger age as compared to those with sporadic NETs [2]. Although NETs are considered rare tumors, their incidence has substantially risen over the last decades (up to 6.98 new cases/100.000 inh/year), mainly due to the increased diagnosis rate of early-stage tumors [3]–[5]. Furthermore, the prevalence of NETs has also significantly increased, thanks to both the higher incidence and the improvement in the treatment armamentarium that prolonged the survival of patients with NET [3].”

Lastly, the correct form for neuroendocrine neoplasms (NEN) has been modified in line 1.

Round 2

Reviewer 1 Report

it is a much more interesting manuscript in its current form 

Reviewer 3 Report

The authors addressed all of my concerns.